# Dynamic Environments with Deformable Objects

**Rika Antonova**[*][§]    **Peiyang Shi**[*][†]    **Hang Yin**[†]    **Zehang Weng**[†]    **Danica Kragic**[†]

[§]Stanford University, CA, USA        [†]KTH, Stockholm, Sweden

## Abstract

We propose a set of environments with dynamic tasks that involve highly deformable topologically non-trivial objects. These environments facilitate easy experimentation: offer fast runtime, support large-scale parallel data generation, are easy to connect to reinforcement learning frameworks with OpenAI Gym API. We offer several types of benchmark tasks with varying levels of complexity, provide variants with procedurally generated cloth objects and randomized material textures. Moreover, we allow users to customize the tasks: import custom objects and textures, adjust size and material properties of deformable objects. We prioritize dynamic aspects of the tasks: forgoing 2D tabletop manipulation in favor of 3D tasks, with gravity and inertia playing a non-negligible role. Such advanced challenges require insights from multiple fields: machine learning and computer vision to process high-dimensional inputs, methods from computer graphics and topology to inspire structured and interpretable representations, insights from robotics to learn advanced control policies. We aim to help researches from these fields contribute their insights and simplify establishing interdisciplinary collaborations.

## 1 Introduction

Deformable objects are ubiquitous in our lives, hence, AI agents and robot assistants would ultimately need to learn to perceive and manipulate such objects. Hand-crafted techniques are likely to fail in unstructured environments, such as households. Therefore, approaches based on machine learning are crucial. Simulation platforms have emerged as powerful tools that help to develop and evaluate learning-based methods using interactive environments [1, 2, 3, 4, 5, 6]. They have the potential to eliminate the shortcomings of training on static datasets [7]. However, most scalable simulation platforms do not provide tasks with deformables. There is significant interest in manipulation of deformable objects in the field of robotics. Nonetheless, due to challenges in perceiving, representing and manipulating deformable objects, a large part of robotics research has focused on tractable tasks. For example, quasi-static tasks to manipulate flat cloth items without fully raising them from the tabletop [8, 9, 10, 11]. Dynamic 'in-the-air' manipulation of highly deformable objects has been attempted, but involved custom high-speed cameras & robot hardware [12] or motion capture for tracking [13]. Nevertheless, customized hardware might not be strictly necessary for success. Recent work showed initial results for dynamic manipulation of cloth [14] and for handling ropes above the tabletop [15]; a recent hardware benchmark outlined off-tabletop manipulation tasks to facilitate robotics research for this direction [16].

Leveraging expertise from multiple research fields would facilitate faster progress for challenging dynamic tasks with highly deformable objects. Machine learning and computer vision researchers could contribute expertise in processing high-dimensional inputs to perceive the state of deformables from images; researchers from computer graphics and topology could develop structured and interpretable representations to improve data efficiency and robustness, robotics researchers could contribute control insights to enable manipulation for advanced tasks.

---

[*]Rika (rika.antonova@stanford.edu) and Peiyang (pyshi@kth.se) contributed equally.

35th Conference on Neural Information Processing Systems (NeurIPS 2021) Track on Datasets and Benchmarks.

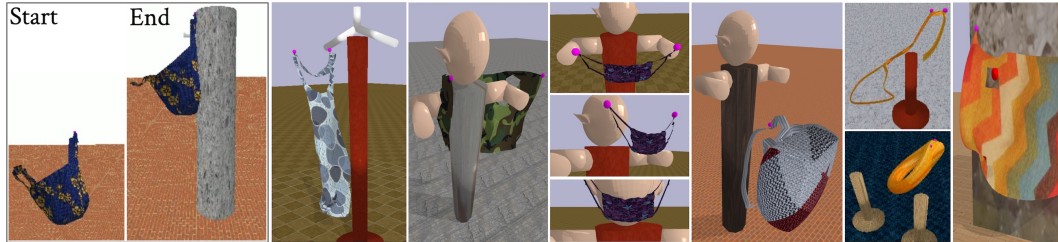

Figure 1: Types of tasks constructed in our simulation environments: hanging various deformable objects onto rigid hooks and hangers; dressing a mannequin, putting masks on a mannequin, putting backpacks on the mannequin's shoulders; throwing hoops or lasso objects onto poles; buttoning tasks that can use procedurally generated cloth with varied hole/button positioning and cloth density.

To enable easy collaboration between various communities, we developed DEDO: Dynamic Environments with Deformable Objects (github.com/contactrika/dedo). We aim for DEDO to be *accessible and user-friendly*:

- fast runtime for large-scale data generation using a free and open-source physics simulator [17]
- easy integration with reinforcement learning libraries using OpenAI Gym format
- TensorBoard integration to view simulated RGB camera images and monitor the training progress
- simplified control interface, grasping via dynamic anchors (users don't need robotics expertise)
- task versions ranging from simple to highly non-trivial, with over 100 new mesh variants
- several procedurally generated tasks to evaluate generalization capabilities of the learning methods
- a variety of textures, variations in material properties and object sizes
- support for loading arbitrary meshes to accommodate experiments with custom deformable objects
- examples of using thousands of meshes from various sources for large-scale unsupervised learning

Our framework relieves users from the need to set up and tune environments with deformables, which is intractable for non-expert users, and is a major roadblock even for experienced researchers [18]. When it comes to simulation with highly deformable objects, especially thin-shell objects, all simulators have significant limitations. Some computer graphics methods can generate realistic visuals, but rendering a few minutes or even seconds of simulation could take hours of compute time. Some physics simulators can achieve physical realism, but do not provide realistic visual output and likewise take hours of compute time even for trivial objects and tasks. We aim to strike a useful balance: faster-than-realtime physics simulation that still offers visualization to support scalable learning from images/videos. Instead of investing considerable computational resources in modeling a particular object or scene precisely, we aim to emulate the overall level of complexity one would face when solving analogous tasks in reality. Hence, our environments should be useful for evaluating the potential of various learning methods quickly, enabling fast iterative research and development.

## 2   Related Work

Promising results have been demonstrated in robotics for learning to manipulate deformable objects, recent surveys give an overview [19, 20, 21, 22]. However, works in this field usually construct a small self-contained scenario to address a specific problem in a constrained lab environment. The setup is usually problem/robot-specific and time-consuming to tune, hence impractical to share with the broader research community. There is long-standing interest in rendering and simulation of deformables in the field of computer graphics [23, 24, 25, 26, 27]. This community emphasizes ability to visualize realistic textures and interactions, but historically has not focused on simulation speed as a main priority. In the field of machine learning, recent success of unsupervised and reinforcement learning methods spurred interest in fast general-purpose simulators. A large number of benchmark environments have been constructed to serve as interesting sources of training data: [28] surveys game environments, [29] lists those for embodied AI (vision and robotics). However, most benchmarks either focus on rigid body simulation [3, 30] or include a single/few tasks that use deformables with simple shapes [31, 32]. SoftGym [33] offers tasks with ropes, liquids and a rectangular cloth object, but adding custom deformables requires re-compilation and a Docker setup for NVIDIA Flex backend [34]. Other recent suites focus on linear objects [35] and plastic materials [36], but lack support for thin-shell deformables, such as cloth and garments. Moreover, [35] uses a paid simulator

Table 1: Task suite comparison. "# Built-in Objects" denotes number of deformable objects included; for DEDO we imported and tested additional 1220 meshes from external sources. (PG) denotes procedurally generated deformables. Customization: (T) customizable texturing; (MP) material properties; (O) load custom deformables ($O^{comp}$ means need to change code or re-compile).

| | Physics Engine | Built-in Object Categories | | | | | Object Variety | |
| --- | --- | --- | --- | --- | --- | --- | --- | --- |
| | | Rope | Cloth | Bag | Garment | Plastic | # Built-in Objects | Customization |
| SoftGym [33] | Flex | ✓ | ✓ | | | | <10 | $O^{comp}$, MP |
| RLBench [32] / Reform [35] | V-REP / AGX | ✓ | | | | | 1 | $O^{comp}$ |
| PlasticineLab [36] | DiffTaichi | | | | | ✓ | <10 | No |
| ThreeDWorld [31] | Flex | | ✓ | | | ✓ | <10 | $O^{comp}$ |
| AssistiveGym [38] | PyBullet | | | | ✓ | | 1 | O |
| DeformableRavens [37] | PyBullet | ✓ | ✓ | ✓ | | | <10 | O |
| DEDO | PyBullet | ✓ | ✓ | ✓ | ✓ | ✓ | 130 + PG | O, MP, T |

(price not listed), [36] requires pre-programmed shape primitives. Hence, there is a need for an easy-to-use and customizable suite with a variety of deformable tasks. Highly deformable objects with non-trivial topologies are of particular interest, since interaction with such objects presents advanced perception and control challenges. Recent versions of PyBullet physics engine [17] support thin-shell objects and easier task construction. [37] used it for tabletop tasks, such as placing cubes into bags and moving cloth rectangles. AssistiveGym [38] includes one assistive dressing task with a hospital gown. Hence, the PyBullet is a good option for building a more extensive task suite with highly deformable objects. Our suite emphasizes support for objects with higher complexity, richer variations and customization. We provide a comparison table (Table 1) highlighting the features of state of the art task suites. PyBullet meets our main criteria: it is free and open source, real-time and scalable; models interactions between rigid and deformable objects; allows loading custom meshes and integrate with python-based deep learning frameworks; offers easy image rendering; is intuitive for users outside robotics, and still offers functionality needed for robotics research.

## 3  DEDO: The Proposed Environments and Tasks

Our Dynamic Environments with Deformable Objects (DEDO) suite includes several classes of tasks: hanging various deformable objects onto rigid hooks and hangers, buttoning with procedurally generated cloth and varied positions for buttons, throwing a lasso rope or hoops onto target poles, putting items onto a mannequin (putting a mask on the mannequin's face/ears, putting a backpack on the mannequin's shoulders, dressing the mannequin). Figure 1 shows several example task instances.

Our main focus is on tasks that are dynamic: objects move in 3D space (instead of resting on a tabletop), gravity and inertia play a significant role. Since the vast majority of existing literature in robotics focuses on quasi-static tabletop tasks, we see the need to turn attention to the challenges of dynamic interactions. We also emphasize working with objects that have non-trivial topology, i.e. contain several loops, openings, holes. This allows to draw interest from communities working on topological representations [39, 40], which are especially promising for highly deformable objects. Topological structures can be used to summarize the global state of a deformable object without the need to keep track of all the local aspects, such as minor folds and cloth wrinkles. Below we describe our tasks and user interface to customize them. All tasks implement OpenAI Gym API [41], with RGB images as observations and velocity commands for grasp anchors as actions.

### 3.1  Hanging Bags and Garments

The objective of the `HangBag` task is to lift a bag from the floor and hang it on a hook (Figure 2). At the start, grasp anchors are attached to a bag handle. We implement grasping as simple anchor attachment. Well-behaved simulation for grasping thin-shell objects is not currently feasible in most simulators, hence we view grasping as a separate line of research that can be addressed by external approaches, e.g. [42, 43, 44, 45]. We also keep the definition of actions simple to avoid requiring extensive control or robotics background from users. The action space is $[-1, 1]^{3 \times 2}$, interpreted as normalized 3D vectors of desired velocities for each of the two anchors. Users don't need to know any further control details. We scale the normalized vectors to ensure that sufficient motion is achieved within task horizon and use a simple proportional controller: applied force is proportional to the difference between current and desired anchor velocity. We use this control strategy for all our tasks.

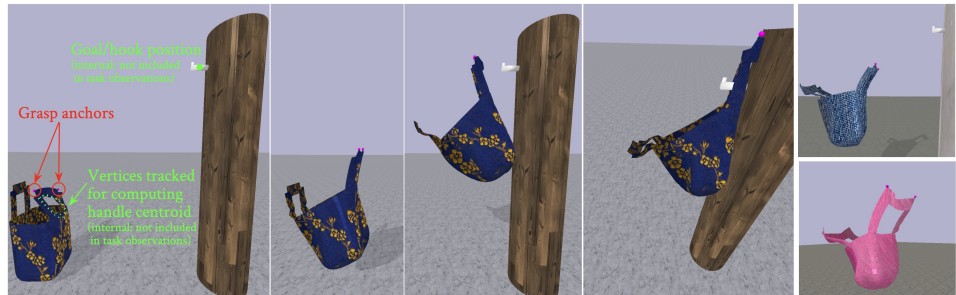

Figure 2: Examples of the `HangBag` task. `HangBag-v1` to `v3` provide task versions with different bag meshes, textures of the floor and pole. `HangBag-v0` randomly selects one of 108 different bag mesh variants, which were generated by reshaping, moving and resizing the bag's main body and handles.

The reward for `HangBag` task is based on how close the centroid of the bag's handle is to the hook: $rwd = -\|centroid(baghandle) - hook_{xyz}\|$. At each step, the centroid of the bag handle is computed by querying the current positions of the mesh vertices of the bag handle, then finding the centroid of the current location for these vertices. This information is only used for computing the reward, and it not exposed as part of observations. At the end of the task the anchors are released and the bag is allowed to drop. If the task is completed successfully, then the bag will rest on the hook and the distance from the bag handle to the hook will remain small. Otherwise, the bag will fall under gravity, the distance will be large, yielding a large negative reward for the final step. Users interested in experimenting with sparse rewards could easily convert this to a sparse reward task by retaining only the final-step reward. We use a similar strategy for computing rewards for all our tasks (please see supplementary material for further discussion).

The objective of the `HangGarment` task is to hang a garment on a hanger. Top row in Figure 3 shows variations of hanging aprons on a hook. The first image also shows the preview of a $400 \times 400$ RGB image used as observations. Users can change the resolution of the RGB images with a flag (e.g. `--cam_resolution=200` to get $200 \times 200$ images). `HangGarment-v6` to `v10` versions load variants of a shirt object to hang. This task is more challenging than hanging an apron, both for perception and control. Bottom row in Figure 3 shows successful and failed attempts. `HangGarment-v0` selects a random garment object to hang and picks random textures for the garment, floor and hanger pole.

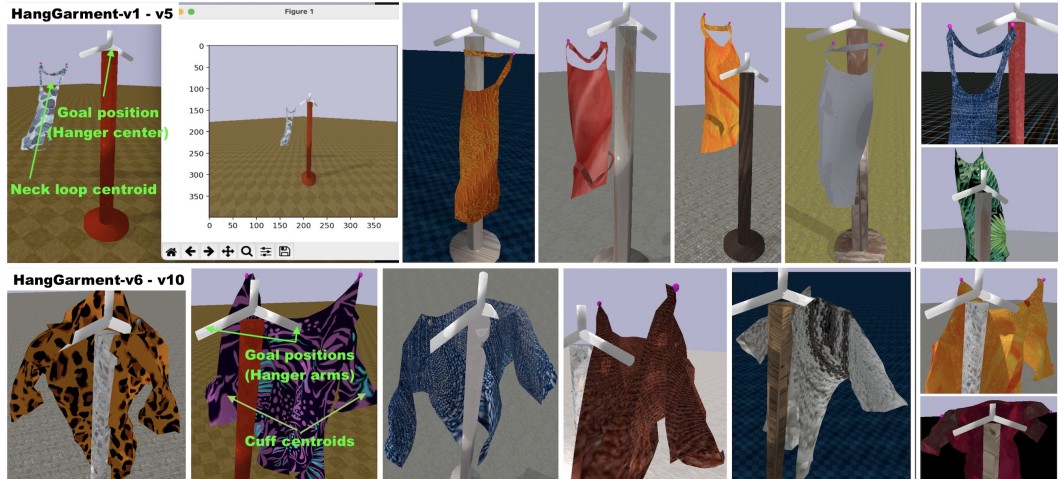

Figure 3: `HangGarment-v1` to `v5`: five aprons with mesh variations and high contrast textures for floor & pole vs apron (top row); `HangGarment-v6` to `v10`: five shirts with mesh and texture variations (bottom row); `HangGarment-v0` loads random garments and textures (last column).

## 3.2 Buttoning with Procedurally Generated Cloth

The objective of the `ButtonProc` task is to align the holes in the cloth with the buttons and pull the anchors so that the cloth comes to rest on the button stems. We use procedural generation for

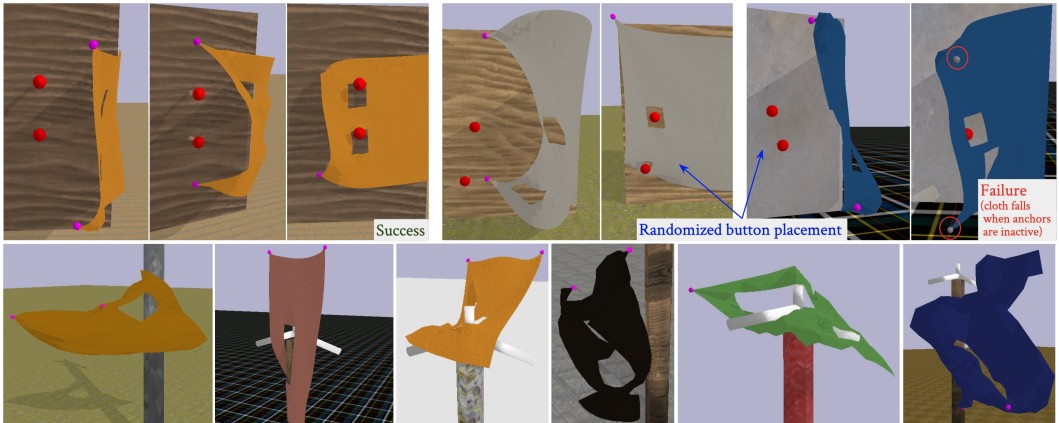

Figure 4: Top row: `ButtonProc` generates cloth on initialization, randomizes positions for holes and buttons. Bottom row: `HangProcCloth` task generates a cloth to hang with one or two holes in random positions; cloth colors, sizes and aspect ratios are also randomized.

cloth meshes, and randomize locations of the holes and buttons. This allows generating an unlimited number of variants for this task. The cloth can be customized to have higher/lower mesh density. The reward is based on the distance between the centroids of the button holes and the buttons.

Procedural generation is important for experimenting with deep learning algorithms, especially unsupervised methods that need a large amount of data to achieve generalization. We designed an additional task with procedurally generated cloth that can be more challenging for perception. `HangProcCloth` generates a cloth with one or two holes whose locations and sizes are chosen randomly. The objective is to hang the cloth using either of the holes. Since the cloth is not attached to anything other than the anchors, its shape changes drastically during the task, hence keeping track of the holes becomes a challenging perception problem.

The tasks we provide can also help to analyze deep learning methods in an interpretable way. For example, if an unsupervised approach such as Variational Autoencoder (VAE) is trained to reconstruct the motion of the procedurally generated cloth with holes, we could inquire whether the latent space of the VAE retains information about the location and sizes of the holes. This would enable insights similar to analysis for rigid objects that can use interpretable low-dimensional states [46].

### 3.3 Throwing Hoops and Lasso

We designed a challenging `Hoop` task with an objective to lift a thick deformable hoop from the first pole and transfer it to the second pole. The reward is based on the distance between the current position of the hoop's centroid and the base of the second pole. Naive approaches are likely to learn to pull the anchors toward the second pole without lifting up. Because the hoop is deformable, such strategy would get an initial increase in rewards, making it easy to get stuck in a local optimum.

Our `Lasso` task provides a further challenge: the task is to throw the lasso onto a pole, with anchors grasping far away from the lasso loop. Holding the lasso far from the loop causes the loop to quickly collapse under gravity. Adjusting the stiffness of the lasso rope gives users ability to vary the complexity of the task. Controlling a stiff lasso would be easy; throwing a highly flexible lasso while maintaining the loop opening is much more challenging, both for perception and for control.

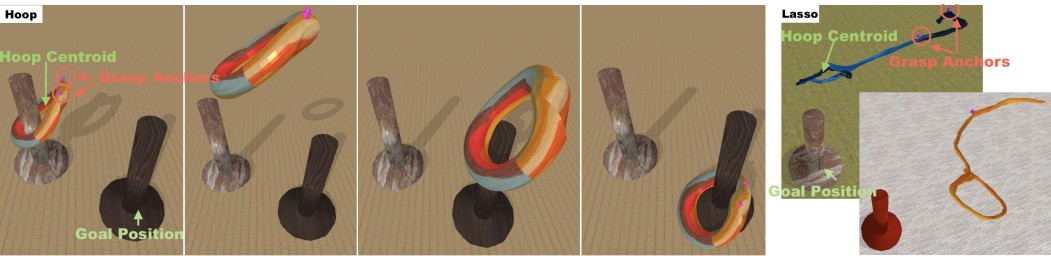

Figure 5: Demo execution of the `Hoop` task (left); examples of images from the `Lasso` task (right).

This task could serve as an interesting test for non-trivial generalization capabilities of the learning methods. For example, users can learn successful policies with a stiff lasso, then evaluate how well the learned networks can be fine-tuned to learn to control a more flexible lasso.

### 3.4 Putting Backpacks and Garments on a Mannequin

We provide a group of tasks to put various items onto a mannequin. AssistiveGym [38] showed that doing a dressing task in PyBullet is possible. However, getting reasonable behavior for something like dragging a long sleeve over an arm requires tuning of the cloth object shape and parameters. Hence, we created several tasks that could be completed without careful tuning, and ensured they are still interesting from the point of view of trajectory generation, geometry and topology of the scene.

The objective of the `Mask` task is to put a mask over the ears of the mannequin. The reward is based on the distance of each mask loop to the corresponding ear of the mannequin. The task starts with the mask in front of the mannequin, thus most algorithms could easily learn to move the mask towards the mannequin's face. However, learning to place the mask loops over the mannequin's ears is significantly more challenging. This is due to partial observability: the motions of the anchors and mask loops would not be visible behind the ears and head of the mannequin. Hence, the learning algorithms would need to infer which policies are successful based on how the front part of the mask moves when the anchors are lowered or released towards the end of the task.

The objective of the `DressBag` task is to put a bag on the mannequin. `DressBag-v1` to `v5` provide task versions for putting backpacks around the mannequin's shoulders. Figure 6 shows a demonstration of successfully completing `DressBag-v1` task, Figure 7 shows variations in the backpack meshes and textures. The variations we provide in `v1-5` versions of our tasks can be used for testing generalization capabilities. For example, learn a policy on 4 out of 5 mesh variations, then test whether it generalizes to a variation and texture that was held out from training.

The objective of the `DressGarment` task is to put a garment on the mannequin. `DressGarment-v1` to `v5` provide tasks to put a vest onto the mannequin. The two holes/loops of the vest are large enough to be clearly visible in the images at the start of the task. However, keeping track of the holes quickly becomes difficult, since the cloth deforms during the task and is obscured by the mannequin, which is in the foreground. This task could be made even more challenging by using shirts with sleeves as garments instead (we provide meshes for several shirt variants).

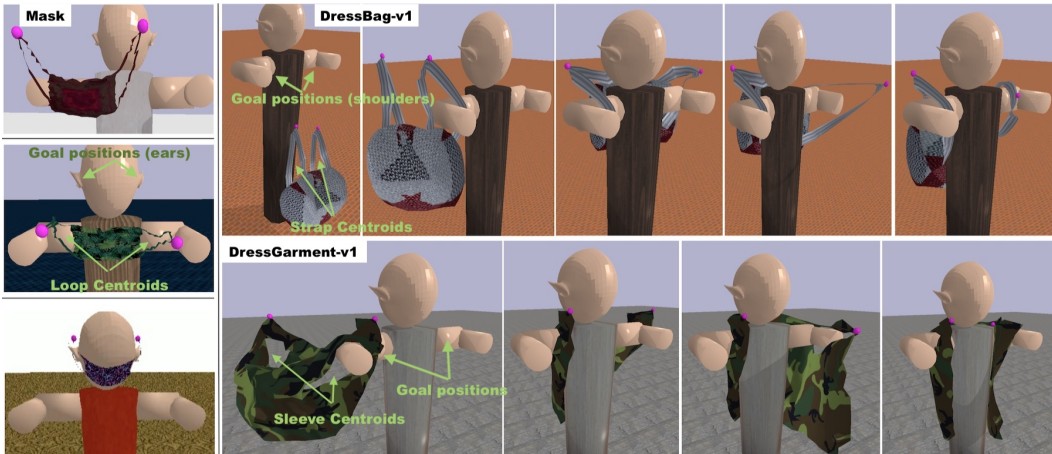

Figure 6: Examples of the `Mask` task (left column); demo of `DressBag-v1` to put a backpack onto the mannequin's shoulders (top row); demo of `DressGarment-v1` to put on a vest (bottom row).

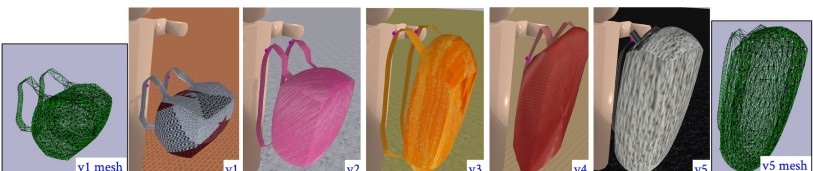

Figure 7: Backpack mesh and texture variations for `DressBag-v1` to `-v5` task versions.

## 3.5 Support for Customization and Extensions

Several recent workshops on simulating, representing and manipulating deformable objects addressed the question during panel discussions of whether standardized benchmarks are desirable [18, 47, 48]. The consensus was that for fields like computer vision and machine learning benchmarks could be quite helpful. However, panelists agreed that for robotics and physical simulation it is more useful to build environments that are easy to customize, so that each lab could create task variants to focus on objects and task aspects they want to consider. Hence, we built our task suite to be customizable.

For each task, users can load a custom `.obj` mesh file via `--deform_obj=[path]`. Users can also specify custom size scaling, damping and bending stiffness, elasticity and friction, provide custom textures for deformable and rigid objects in the scene. `--cam_resolution` flag allows to specify custom size of the RGB images, users can also specify custom camera angle and positioning.

To demonstrate how meshes from external datasets could be easily incorporated into our environments, we created `Sewing` and `BGarments` examples. `Sewing-v0` selects random garment meshes out of 1200 meshes from the Sewing Patterns dataset [49]. We applied basic mesh simplification to retain ≈2000 vertices per mesh. The Sewing Patterns dataset contains more than 22000 garment variations that are procedurally generated from sewing templates. Users could import all meshes from this dataset, if they would like to benefit from this diversity. `BGarments-v0` provides another example of importing external meshes: loads meshes from [50] and applies a random texture.

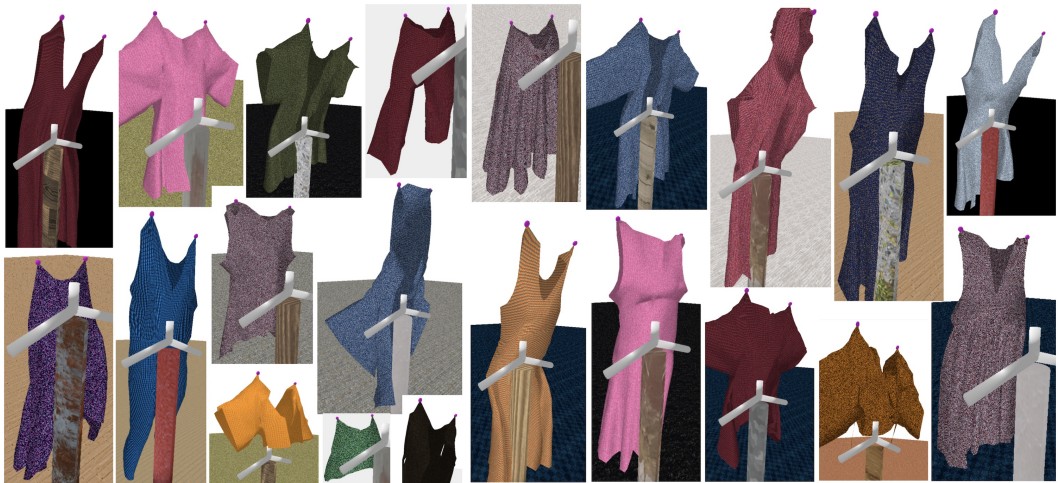

Figure 8: `Sewing-v0` randomly selects one of 1200 meshes we imported from the Sewing Patterns dataset [49], which has over 22000 meshes total (all can be easily imported).

## 3.6 Experimental Robot Manipulation Tasks

DEDO aims to support interdisciplinary research, giving a simple gripper anchor interface that abstracts away robot control specifics. Nonetheless, for robotics researchers we added the widely used Franka Emika robots to create dual-arm manipulation scenarios. Figure 9 shows an example.

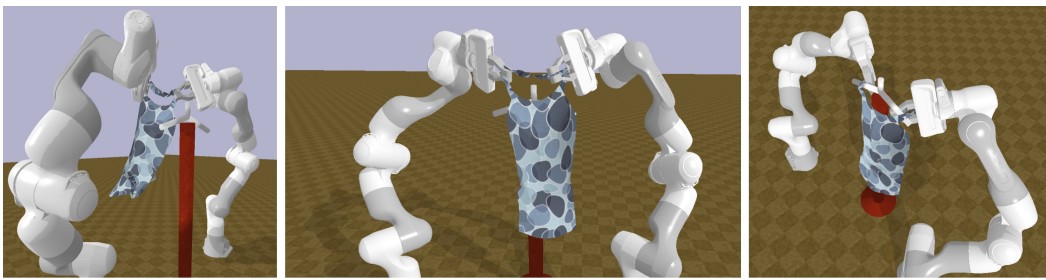

Figure 9: An example of dual-arm manipulation with Franka Emika robots for `HangGarment` task.

# 4 Unsupervised Learning and Reinforcement Learning with DEDO

The problem of perceiving and representing the state of highly deformable objects presents an excellent opportunity for unsupervised learning and reinforcement learning to showcase their potential. We provide several examples of how approaches from these fields can be run on our environments.

## 4.1 Example Integration with Unsupervised Learning

To encourage experiments with unsupervised approaches we show examples with several widely used and recently proposed Variational Autoencoder (VAE) variants:

- $VAE$: based on the original VAE from [51], uses a convolutional encoder and the corresponding de-convolutional decoder (this conv-deconv stack is also used for the VAE-based methods below).
- $SVAE$: a sequential VAE that reconstructs a sequence of images $x_1, ..., x_t$; the output of the convolutional layers goes through an additional LSTM layer before decoding. Actions $a_1, ..., a_t$ constitute additional input when doing reconstructions (for this and all other sequential variants).
- $PRED$: a VAE that takes a sequence of images $x_1, ..., x_t$ and learns to output a predictive sequence $x_1, ..., x_{t+k}$. This variant was shown to perform well in tasks with rigid objects [46].
- $DSA$: Disentangled Sequential Autoencoder [52] that separates static vs dynamic aspects of latent state and learns the latent dynamics; uses bidirectional LSTMs in static and dynamic encoders.

Figure 10 shows a visualization of our TensorBoard logging. In addition to common training metrics in the `SCALARS` tab, we also provide visualizations of the reconstructions after each training epoch in the `IMAGES` tab. Training runs are given descriptive names based on the learning algorithm, start date/time, environment/task name and version. `TEXT` tab shows command-line flags. Our example training scripts also allow to log to `wandb.ai` service, which provides further customization for plots.

We trained $VAE, SVAE, PRED, DSA$ on several tasks from our suite (`HangBag, HangGarment, Button, BGarments, Sewing`). Training ran on streaming data obtained by executing a random policy to get sequences of RGB images as training batches. Surprisingly, $VAE$ outperformed the more advanced methods. The sequential VAE variants ($SVAE, PRED, DSA$) did show stronger performance in environments with rigid objects. It is likely that learning structured latent states from simulations that involved only rigid objects was much more tractable than dealing with highly deformable objects. This indicates the need for further research with tasks that include deformables.

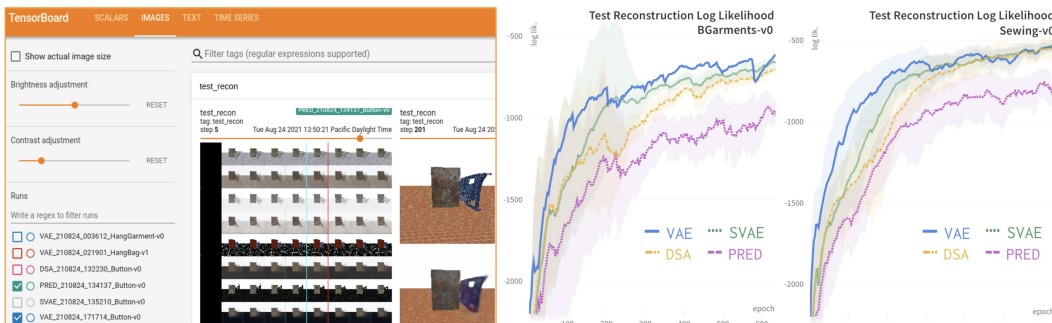

Figure 10: Left: our TensorBoard visualizations showing reconstructions for `Button-v0`. The two right plots show VAE results on `BGarments-v0` and `Sewing-v0`: mean of 5 seeds/runs for each curve, shaded regions indicate one standard deviation; for each epoch we get 400 new frames; we sub-sample minibatches for training iterations s.t. 100 epochs take ≈1 hour for each VAE variant.

## 4.2 Reinforcement Learning with Hyperparameter Search

To facilitate experiments with reinforcement learning (RL) we included examples of using RL frameworks. One widely used and user-friendly library that supports a variety of RL algorithms is StableBaselines3 (SB3) [53]. We integrated SB3 with our TensorBoard video logging. Videos (that we log to `IMAGES` tab in TensorBoard) are useful for qualitative evaluation of RL training, as a complement to quantitative evaluation with reward statistics and neural network loss plots (in `SCALARS` tab in TensorBoard). Training speed on our tasks ranges from 120 FPS (`HangProcCloth`)

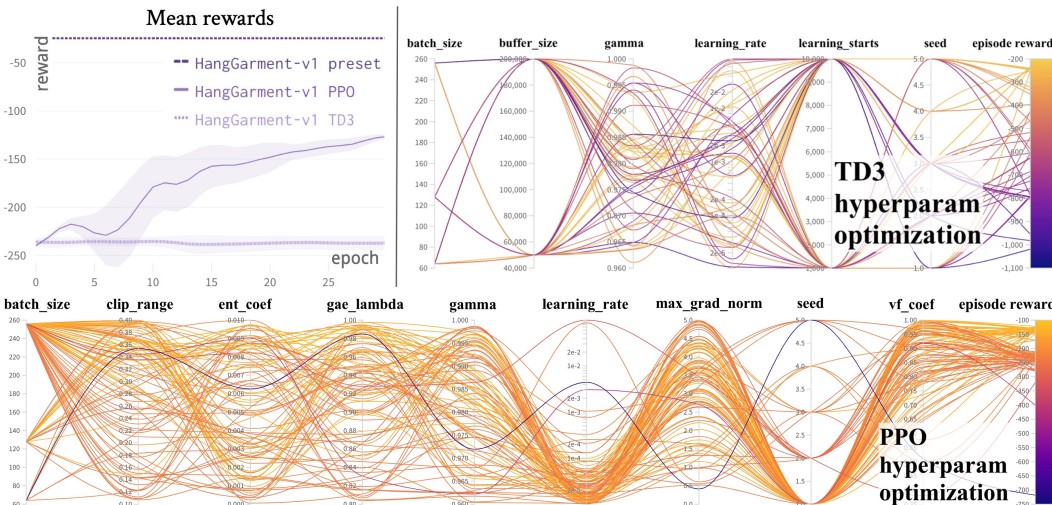

Figure 11: The first plot shows rewards for 'preset' vs RL policies: mean of top 5 runs for PPO/TD3; shaded regions show one standard deviation; each epoch gets ≈16K environment steps/frames (from 8 parallel environments). The next two plots visualize hyperparameter search for `HangGarment-v1`.

to 30 FPS (`DressGarment`) on a single NVIDIA GTX 2080 GPU and 8 CPU cores, including simulation an RL training time. We also added support for RLlib [54] – a scalable library with over 30 RL algorithms that offers multi-GPU training and distributed data collection from actors using CPUs on different machines.

We trained several RL algorithms on our tasks: `A2C, PPO, SAC, TD3`. Of these, only `PPO` [55] was able to make progress on most tasks without hyperparameter tuning. PPO also showed versatility on another recently released suite – NVIDIA IsaacGym [5]. PPO learned successfully on a variety of tasks with rigid objects in IsaacGym. However, PPO policies only moved deformables closer to the goal regions in DEDO tasks, but did not end up solving the tasks without hyperparameter tuning.

To investigate whether hyperparameter search could help RL, we selected one on-policy algorithm (PPO) and one off-policy algorithm (TD3) and performed Bayesian hyperparameter search with Hyperband [56]. For each task + RL algorithm, we allocated 8 Intel CPUs (Xeon 4110 2.10Ghz), 30GB RAM, and 1 NVidia RTX2080Ti GPU per task, totalling 7 CPU-compute days for each task + RL algorithm pair. We also ran a more extensive optimization for `HangGarment-v1` with additional 5 CPU-compute days for PPO and TD3 (each).

For our experiments we used dense rewards (closeness-to-goal-region described in Section 3) combined with the final-step reward: after the last action the anchors were released, and if objects were not hanged/hooked/put on well – a penalty was added. This penalty indicated that actually completing the task was a lot more valuable than simply moving towards the goal region. The penalty was computed in a similar way for all tasks (supplementary provides further details). Despite the extensive hyperparameter search, PPO showed similar qualitative behavior as with default parameters: for `HangGarment-v1` it only learned to bring the deformable object closer to the goal region without solving the task. These results are visualized in Figures 11 and 12.

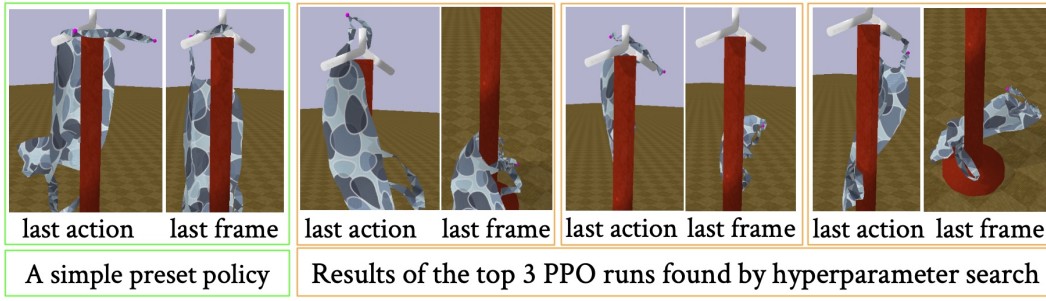

Figure 12: Qualitative results – PPO learns to bring the garment towards the pole ('last action' is shown), but does not learn to hang – the garment falls after the anchors are released ('last frame').

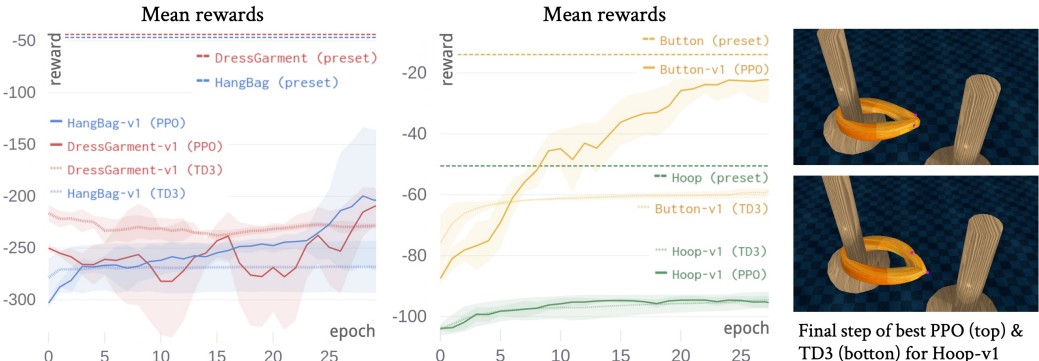

Figure 13: Left & middle: rewards for RL with hyperparameter search (same format as in Figure 11) for HangBag, DressGarment, Button & Hoop tasks. Right: qualitative results for the Hoop task.

The first plot in Figure 13 shows results for HangBag and DressGarment tasks. For HangBag: learning to approach the hook was tractable. However, hanging objects such that they don't fall was more difficult. This is clear from the large gap between the reward of a simple preset policy vs PPO/TD3. The second plot shows learning on Button-v1 and Hoop-v1 tasks. The last column visualizes the best PPO and TD3 policy on Hoop-v1. Both learn to pull the hoop towards the other pole, but fail to learn lifting it up to transfer to the other pole.

Overall, PPO performed well on Button-v1 task, however, for all other tasks RL algorithms made initial progress, but struggled to solve the tasks. This indicates that DEDO tasks present a formidable challenge to the current model-free RL algorithms. Hence, it could be interesting to explore using additional information, such as demonstrations. We provide preset demonstrations of how each task can be solved by controlling grasp anchors through a sequence of waypoints. Some of these trajectories are visualized in Figures 5 & 6. Randomizing these waypoints can yield a set of initial demonstration trajectories for 'learning from demonstration' methods.

## 5   Discussion and Future Work

We developed a set of environments with highly deformable objects that can serve as a source of challenging data. We prioritized flexibility and support for customization, allowing users to load numerous custom meshes, either from their own scans or from external sources. Our environments use free and open source physics simulator, eliminating the need to purchase simulation licenses. The suite is easy to setup with a just simple 'pip install' command, making it accessible for researchers without prior experience with physics simulation. At the same time, the simulation engine can accommodate expert robotics users: custom robot models can be loaded, advanced robot control techniques could be used instead of the simplified anchor interface we provided for novice users.

Our current choice of the physics engine (PyBullet) does prioritize speed over visual realism. Hence, we track advances in simulators and would diversify to support new engines that could achieve good speed-vs-realism balance in the future. For advanced use cases in robotics, it would be interesting to add mobile manipulators to the scenes and experiment with household tasks, such as collecting clothes for laundry, cleaning food from plates, assisting with dressing tasks. This extension would help to evaluate robotics algorithms for their potential to tackle a variety of real world tasks.

**Ethical Concerns and Broader Impacts**: This work contributes a suite of environments that aims to help further research on representing and manipulating deformable objects. Since this is a challenging area, in the short term we do not foresee any broad societal effects, since the results of the works that use this suite would be limited to academic research. If works like this bring more attention to the field in general, then there could be several long-term effects. The positive effects would be progress in robotics for handling everyday household objects, including deformables. Certain industry sectors could benefit as well, such as textile manufacturing, recycling, and hospital care. Such long-terms effects could eventually result in job displacement, so it would be important to create new roles and jobs for people. For example: develop user-friendly interfaces for managing robot assistants, so that people could be seen as valuable employees even in industries with high levels of automation.

## 6   Acknowledgements

Rika Antonova is supported by the National Science Foundation grant No.2030859 to the Computing Research Association for the CIFellows Project. Peiyang Shi, Hang Yin, Zehang Weng and Danica Kragic acknowledge the sponsorship from the Swedish Research Council, Knut and Alice Wallenberg Foundation and European Research Council (BIRD project). Hang Yin would also like to thank the Swedish Promobilia Foundation.

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
