# Supplementary Material for
# Dynamic Environments with Deformable Objects

**Rika Antonova**[*§]    **Peiyang Shi**[*†]    **Hang Yin**[†]    **Zehang Weng**[†]    **Danica Kragic**[†]

[§]Stanford University, CA, USA        [†]KTH, Stockholm, Sweden

## Contents

## 1  Links and Documentation: Items Required by the NeurIPS Instructions

1. *Dataset documentation*: DEDO repository github.com/contactrika/dedo contains the README, which serves as a user guide; the Wiki at github.com/contactrika/dedo/wiki contains architecture overview and further details.

2. *Statement of responsibility*: We (the authors) agree to bear all responsibility in case of violation of rights, etc., and confirmation of the data license.

3. *Hosting plan*: Our code and assets are hosted on `github.com`.

4. *Links to access the dataset and its metadata; openness; long-term preservation; license; metadata; DOI*: DEDO repository is at github.com/contactrika/dedo under MIT license. DOI: 10.5281/zenodo.5838440

5. *Reproducibility*: Our repository includes all the training scripts that we used to produce evaluations for the paper; the README in our repository contains instructions for installation, training and visualizing the results.

## 2  Architecture Overview

Since we provide a customizable simulation suite, we wrote documentation to guide the users through the available tasks, features and customization options. To avoid duplicating it here, in the supplementary we only include a screenshot with the overview of the DEDO architecture (Figure 1).

The core of our suite is the DEDO environment (`DeformEnv` in `dedo/envs/deform_env.py`). This class is served by various utilities that help loading built-in and custom meshes, running procedural

35th Conference on Neural Information Processing Systems (NeurIPS 2021) Track on Datasets and Benchmarks.

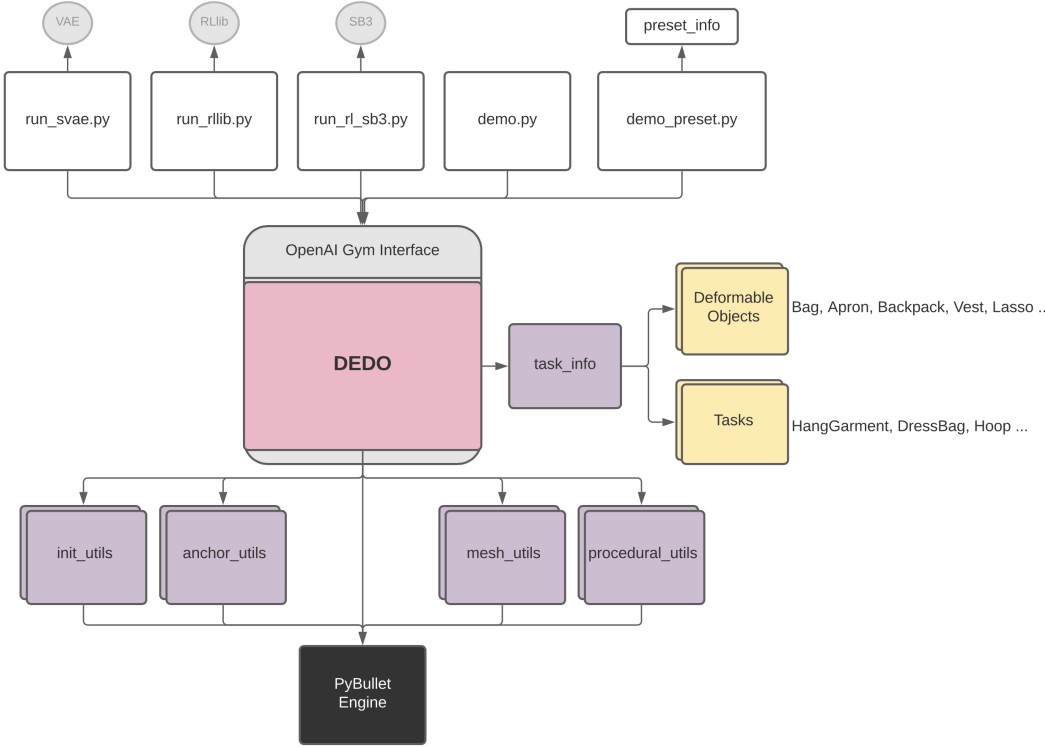

Figure 1: DEDO architecture overview from our documentation page.

cloth generation, initializing and updating the physics engine, etc. The top-level part that most users will interact with are the example demo and training scripts. `demo.py` gives a succinct example of loading the tasks and running a simple policy, without any further dependencies / models. `demo_preset.py` shows how the tasks can be solved by following a set of pre-defined waypoints. `run_rllib.py` and `run_rl_sb3.py` launch RL training, `run_svae.py` launches the SVAE training. In our documentation we have included detailed description of the arguments that these user-facing scripts accept. For users who would like to extend our environments we provided an explanation of how to modify existing tasks and add new custom tasks.

## 3   Reward Function Details for Tasks

The reward functions are all based on the distance between a centroid of a deformable loop to a goal position on the rigid body. For tasks that involve hanging/dressing/buttoning, at the end of each task, anchors are released to let objects free fall. If the task is completed successfully, the deformable object will be supported by the rigid body (e.g. a hanger supporting an apron). For Lasso and Hoop, an outward pulling force is applied to the deformable object. This is to ensure the object's center loop is placed through the target/goal pole, rather than hovering/wrapped around it.

**HangBag**, **HangGarment**, **Hoop**, **Lasso** – these tasks have a single goal position and a single annotated loop. This 'ground truth' annotation indicates which vertices of the deformable object are on the loop. This information is only used to compute the reward and is not visible to the learning algorithms. We compute reward based on the distance of the centroid of the loop to the goal position:

$$rwd = -\|centroid(loop) - goal_{xyz}\|.$$

For `HangBag`, the loop is the bag handle, the goal is the hook. For `HangGarment`, the loop is the apron's neck loop, the goal is the hanger. For `Lasso`, the loop is the center of the lasso loop, the goal is the bottom of the pole. For `Hoop`, the loop is the center of the hoop; the hoop is initialized to rest at the bottom of one of 2 poles, and the goal is the center of the 2nd (the other) pole.

**DressBag**, **DressGarment**, **Button**, **ButtonProc**, **Mask** – these tasks have two annotated loops and two matching goal positions. The reward is the sum of distances from each of the loop's centroid to their respective goal positions:

$$rwd = -\left(\|centroid(loop1) - goal1_{xyz}\| + \|centroid(loop2) - goal2_{xyz}\|\right).$$

For `DressBag` and `DressGarment` tasks, the loops are the the two shoulder straps of the backpack and the two sleeve holes on the vest. Both have a goal position near the shoulder region of the humanoid figure. For `Button` and `ButtonProc` tasks, the loops are the two openings on the cloth; the goal positions are below the buttons. For `Mask` task, the loops are the mask's ear loops and the goal positions are behind the ears of the humanoid figure.

**HangClothProc** – this task has procedurally generated cloth, with up to 2 holes, which are cut out randomly. The reward is computed based on the distance of the closest cloth hole/opening to the center of the hanger (goal):

$$rwd = -\min\left[\|centroid(procHole1) - goal_{xyz}\|, \|centroid(procHole2) - goal_{xyz}\|\right].$$

**Combined/Episode Rewards** – for our RL experiments, we used dense rewards described above combined with the final-step reward: after the last action the anchors were released, and if objects were not hanged/hooked/put on well – a penalty was added. This penalty indicated that actually completing the task was a lot more valuable than simply moving towards the goal region. Concretely, the final-step reward was: $rwd_{final} = -\|centroid(deformable\_loop) - goal_{xyz}\| \cdot MULT$, where $MULT = 400$. Our tasks had episode lengths of 200-700 steps, so a multiplier of 400 was chosen to make the final-step reward be on par with the cumulative dense reward for the episode. In our earlier experiments we also tried $MULT = 50$, which yielded similar qualitative RL results.

## 4 Food Packing Task as an Example of Advanced Customization

DEDO framework can be quickly adapted to construct other tasks that might be of interest to vision or robotics communities. We demonstrate this with an example of how to make a `FoodPacking` task. The goal in this task is to move various food items into a small packing area. The food items include rigid objects, such as tin cans and packaged food boxes, as well as deformable objects, such as fruit. For this illustration we imported object meshes from the YCB dataset [1], which is a widely used dataset in robotics, since research groups can obtain the real counter-

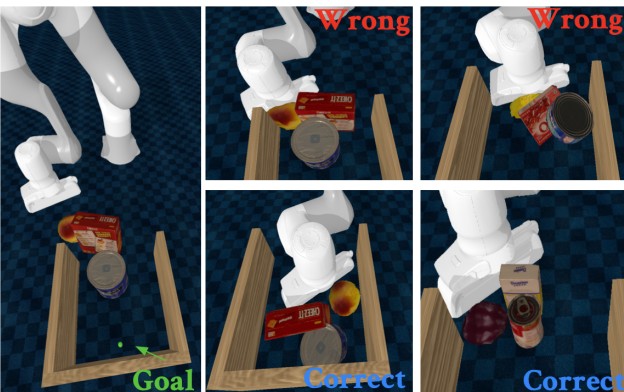

Figure 2: Examples of running a `FoodPacking` task.

parts of the simulated objects. These meshes and their textures were scanned from real objects. We first down-sample the meshes to contain $\approx 1000$ vertices and load them into PyBullet. We then define the reward for the task to be a combination of how close the objects are to the goal region and how much the deformable fruit objects deform. Figure 4 illustrates the task setup. We also illustrate executing several preset robot trajectories, the ones labeled 'Wrong' squish the deformable fruit objects, while those labeled 'Correct' show an example of how the robot can move the objects closer to the goal region without causing large deformation of the fruit object.

We did not consider tasks such as food packing as core tasks for the DEDO suite, since the primary goal of DEDO is to provide a suite of dynamic tasks with highly deformable objects that have interesting topologies. Nonetheless, we wanted to give potential users further ideas for tasks and other research directions that DEDO extensions could facilitate.

## 5 Additional Hyperparameter Search Plots

Below we show figures visualizing our hyperparameter sweeps for PPO and TD3 RL algorithms.

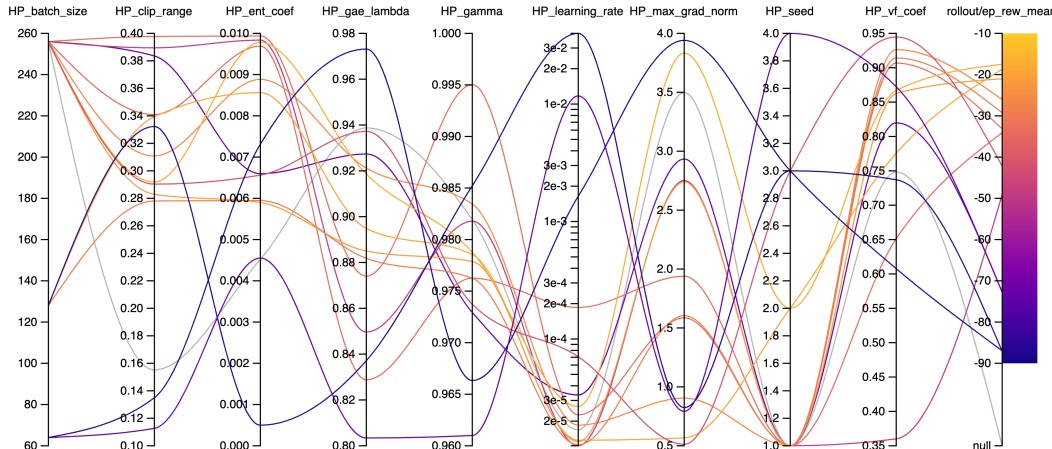

Figure 3: Button-v1 - PPO

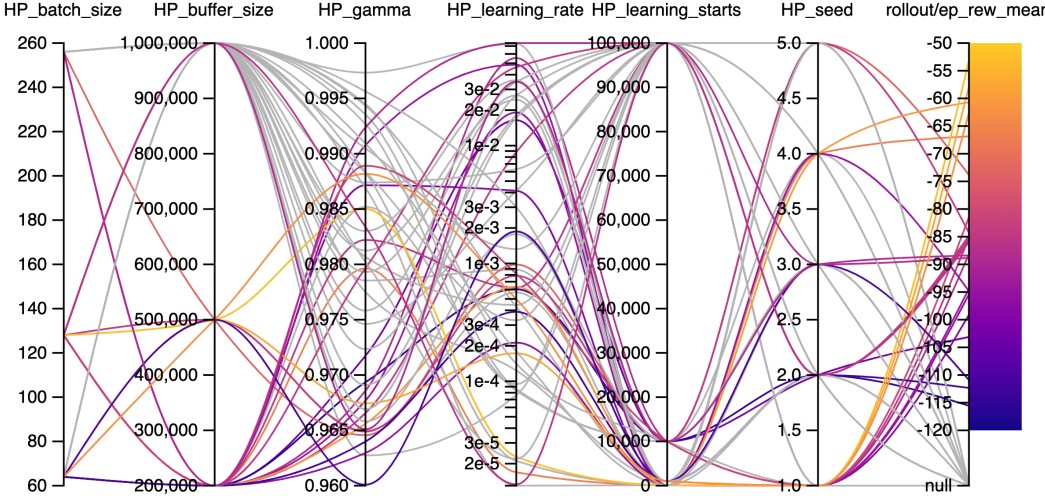

Figure 4: Button-v1 - TD3

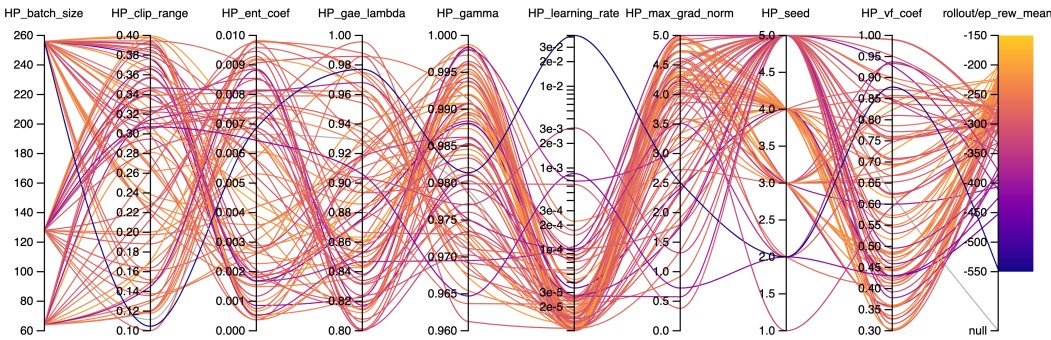

Figure 5: DressGarment-v1 - PPO

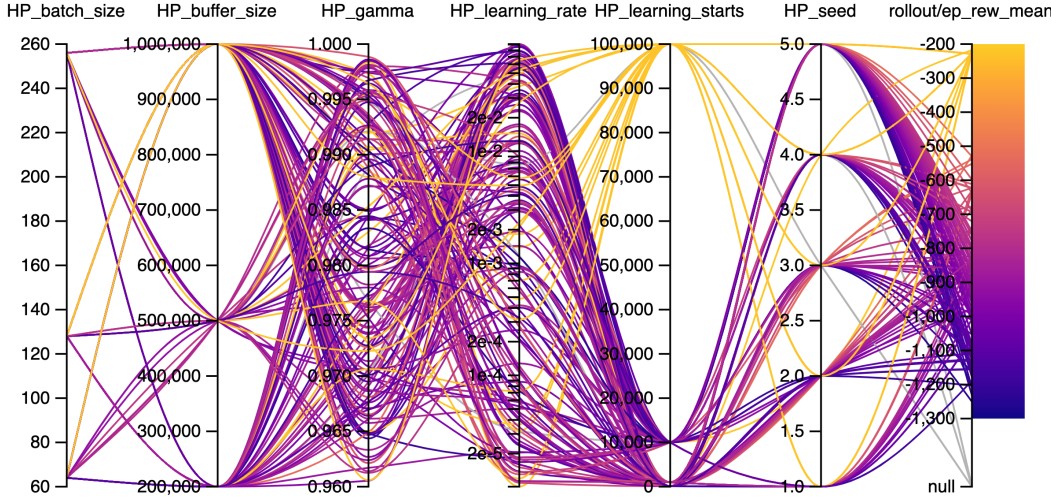

Figure 6: DressGarment-v1 - TD3

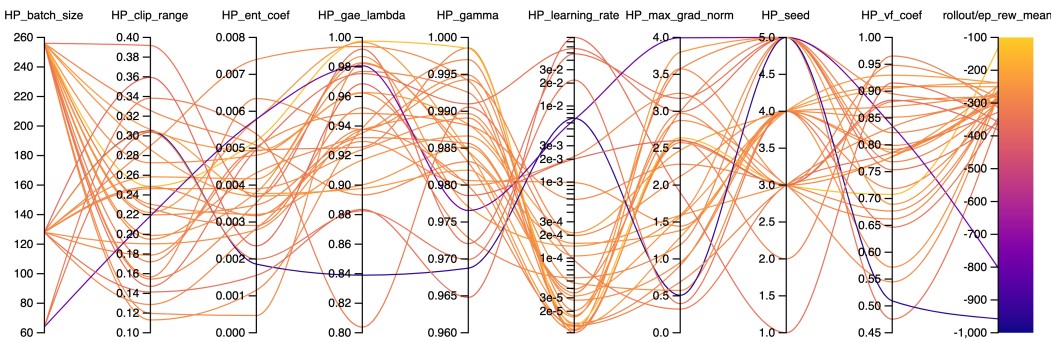

Figure 7: HangBag-v1 - PPO

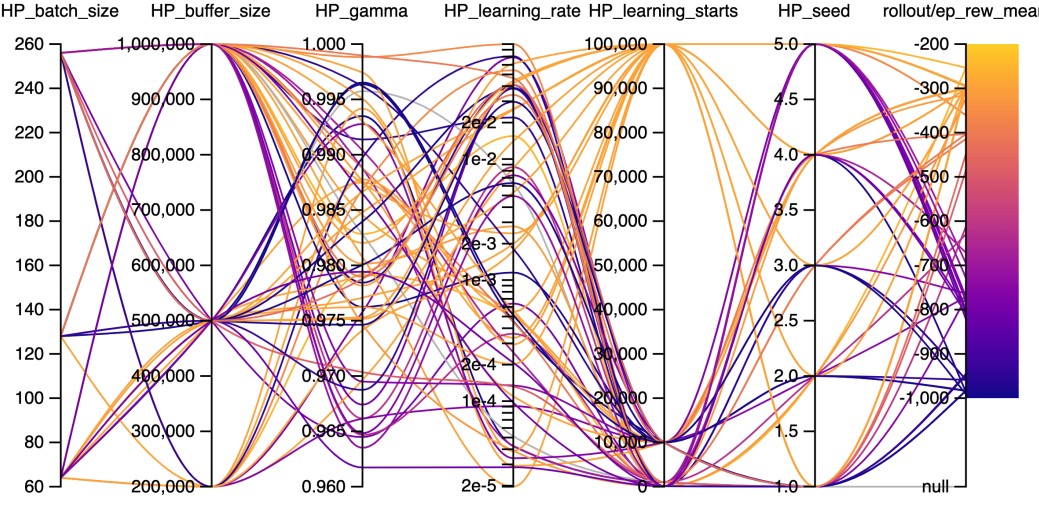

Figure 8: HangBag-v1 - TD3

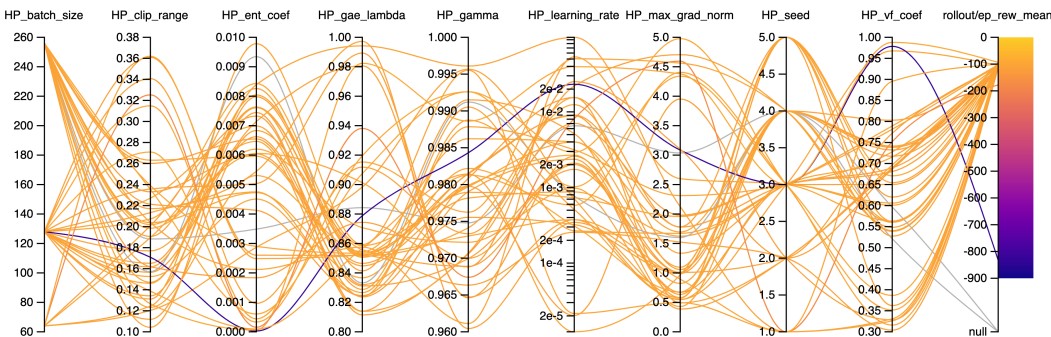

Figure 9: Hoop-v1 - PPO

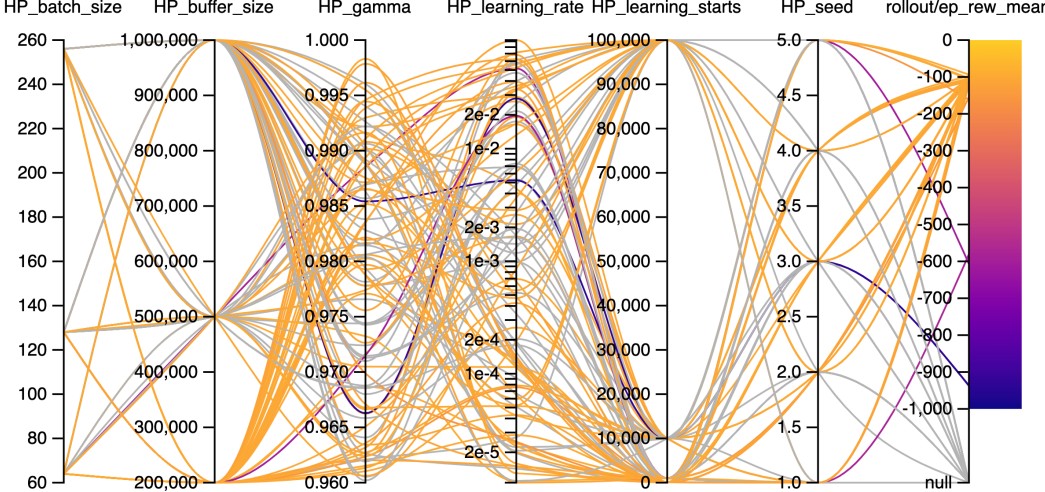

Figure 10: Hoop-v1 - TD3

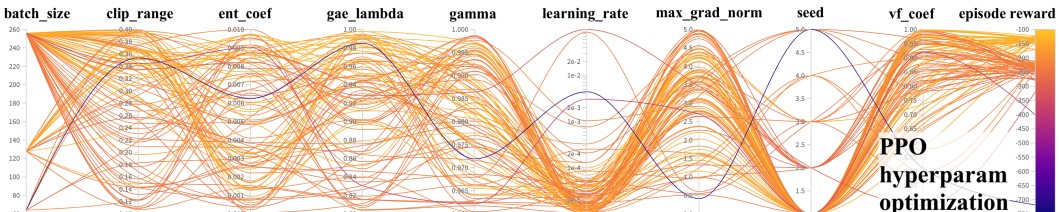

Figure 11: HangGarment-v1 - PPO (a larger version of the plot from the main paper).

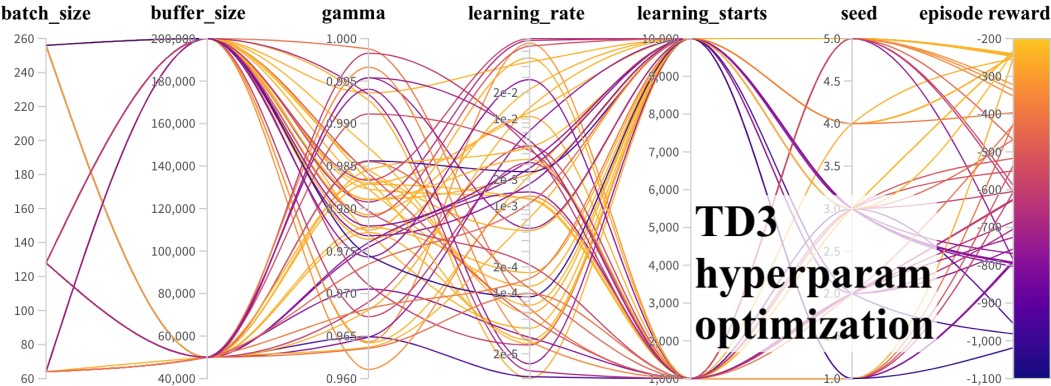

Figure 12: HangGarment-v1 - TD3 (a larger version of the plot from the main paper).


## References

[1] Berk Calli, Arjun Singh, Aaron Walsman, Siddhartha Srinivasa, Pieter Abbeel, and Aaron M Dollar. The YCB object and model set: Towards common benchmarks for manipulation research. In *International Conference on Advanced Robotics (ICAR)*, pages 510–517. IEEE, 2015.