# OpenReview forum: "Dynamic Environments with Deformable Objects"
_NeurIPS.cc/2021/Track/Datasets_and_Benchmarks/Round2 — NeurIPS 2021 Datasets and Benchmarks Track (Round 2)_

### Official Review · Reviewer_gxYx · 2021-09-16
**DEDO Review**

**Rating:** 6
**Confidence:** 4

**Strengths:**

* Manipulation of deformable/cloth objects is an interesting and underexplored topic.
* The code for this benchmark is well-organized and functional
* The authors use a free and open-source physics simulator

**Weaknesses:**

* It seems that there are already existing RL benchmarks for deformable objects. This benchmark has some more varied objects such as garments, but it's not clear what interesting challenges these novel objects pose to RL algorithms that aren't already in existing deformable benchmarks such as SoftGym.
* Authors claim that their platform makes it easier to add custom objects (you have to recompile SoftGym), but this seems like a minor technical difference.
* The authors make the statement, "It is likely that learning structured latent states from simulations that involved only rigid objects was much more tractable than dealing with highly deformable objects. This indicates the need for further research with tasks that include deformables". While highlights an interesting property of unsupervised learning approaches, this conclusion could just have easily been made from existing benchmarks for deformable objects and is not specific to this set of deformable objects.
* More could be done to make these tasks resemble real-world robotics tasks. An environment with floating fingers at prespecified grasp anchors seems too oversimplified. This concern appears to be justified by how easily PPO can solve these tasks with almost no variability.
* Even though each instance of a task has different garments of the same class (e.g. bags with different textures), it's unclear how much this affects the policy. Looking at the learning curves, it appears the specific garment doesn't affect the policy at all. There should be more variance in different instances of a task that demonstrates the necessity of using the observations.
* In the conclusion section, the authors state that they "plan to add mobile manipulators to the scenes and experiment with household tasks." This addition would certainly go a long way to make the tasks more interesting, challenging, and transferrable to real-world domains.

**Additional Feedback:**

While this is an interesting direction, the tasks in this work don't present any unique difficulties not present in other deformable benchmarks, and the learned policies are not transferable to real-world robotics domains. In addition, the evaluation is not sufficient as the paper lacks many task evaluations, baselines, and experimental details.

**Clarity:**

* The hyperparameters used for the RL algorithms are not specified in the text.
* The architectural details for any of the evaluations (unsupervised or RL) are also not described in the text.
* If more room is needed for experimental details and further evaluation, it seems like many of the figures are redundant/unnecessary.

**Correctness:**

* Why was no hyperparameter search done for the various RL algorithms? I think this is common practice for benchmarks to compare algorithms fairly by doing a hyperparameter search for each algorithm instead of simply using the defaults.
* What do the error bars on the plots mean?
* Even though authors supposedly tested on multiple RL algorithms, only results for a single RL algorithm (PPO) are shown.
* Results for the lasso and buttoning tasks are not shown even though those seem to be the most difficult tasks.
* Only the results for two of the tasks are shown in the unsupervised learning setting.

**Documentation:**

The documentation and codebase appear to be well-organized and functional.

**Relation To Prior Work:**

All related work work is discussed.

**Summary And Contributions:**

Dynamic Environments with Deformable Objects (DEDO) is a set of tasks that involve deformable objects with special attention paid to everyday deformable objects such as masks, bags, clothes, and rope. The environments are implemented in PyBullet and use a publicly available repository of garment object models but are easily extensible to custom object models. A subset of the tasks is evaluated in unsupervised learning and reinforcement learning settings.

---

> ### Author Response · Authors · 2021-09-29
> **We have now completed an extensive hyperparameter search for RL PPO/TD3 and clarified that most tasks are challenging (not solved)**
>
> Overall, we thank the reviewer for noting that the complexity of manipulation with highly deformable objects was not made sufficiently clear in our original submission. Having worked in this field for several years, we got used to the fact that both simulation and hardware/real-world aspects are usually painfully difficult. But we missed the fact that this should be explained further and made explicitly clear.
>
> > Looking at the learning curves, it appears the specific garment doesn't affect the policy at all
> > This concern appears to be justified by how easily PPO can solve these tasks with almost no variability.
>
> We thank the reviewer for showing we did not communicate clearly that these tasks were very difficult for RL. We thought that the readers would look at the saturation in the learning curves and conclude that the RL policies were not able to solve most of the tasks. The performance with small variability indicates that the policy easily learned to move deformables closer to the target regions; however, it consistently failed to ensure the desired topological relations (e.g. putting sleeves/strap/loops onto hands/handles/hoops). Unfortunately, with the plots and reward scaling in our original submission it was difficult to see this point clearly. Moreover, since we did not initially do an extensive hyperparameter search, one could wonder whether RL performance was limited by a default choice of hyperparameters.
> Hence, we took the time now to do an extensive hyperparameter search with a state-of-the-art Bayesian approach, described the main insights in the new 10th page of the paper, and included more plots in the supplementary. When the reward is structured s.t. failure to solve a task is appropriately penalized, it is now evident in the plots that PPO cannot solve most of the tasks in our suite. We also included qualitative results showing the state at the end of the episode for the top-performing PPO and TD3 runs found by the hyperparameter search (Figure 13 on the new page 10). We spent a significant amount of compute resources to conduct hyperparameter search (details are given on page 10), and thus hope that we were able to clarify this point sufficiently.
> We also now clarified the range of difficulty of the tasks in Figure 2 of the supplementary pdf. There, we show the performance of the top3 PPO and TD3 policies found by the hyperparameter search on an easy task (Buttoning-v1) and a difficult task (DressGarment-v1). PPO solves Buttoning-v1, TD3 is able to make some progress but lags far behind -- hence this is an example of an ‘easy’ task. In contrast, both PPO and TD3 fail to make progress on DressGarment-v1 task. ‘-v0’ versions of the tasks would be more difficult that our ‘-v[1+]’ versions, since ‘-v0’ versions use randomized textures and mesh variants. To put the reward curves in context - we plotted the episode reward obtained by simple hand-constructed trajectories that do solve the tasks. These trajectories are visualized in the illustrations in our paper and in the supplementary video + README. We also provided example snapshots of hyperparameter searches we conducted for PPO and TD3 that show the range of hyperparameters we considered and the resulting performance/reward distributions.
>
> Related to the above:
>
> > Even though authors supposedly tested on multiple RL algorithms, only results for a single RL algorithm (PPO) are shown.
> > Results for the lasso and buttoning tasks are not shown even though those seem to be the most difficult tasks.
>
> Only PPO showed consistent progress. A2C, TD3, SAC (from StablesBaselines3), DDPG, Impala (from RLlib) were not able to make consistent progress in 500K environment steps with default hyperparameters. After getting a flat line in a few initial runs we did not come back to tuning these. We did not have compute capacity to run an extensive hyperparameter search on all of these, so chose one on-policy and one off-policy algorithm for a more in-depth analysis. Buttoning-v1 was actually easy with the current setup (Figure 2 now in supplementary). We did not see any progress on solving the Lasso task, so we focused on the Hoop task instead. While we did invest effort in addressing RL-related comments, we do not aim for DEDO to become an RL-centric benchmark. DEDO is aimed to help the community to make progress on representing and manipulating deformables. We are not advocating that this should be only done with RL or that RL should be evaluated using DEDO. Hence, we did not aim to provide an exhaustive set of statistics for RL. Our experiments were aimed to show RL can make some progress, but the tasks remain far from being solved.
>
> > The hyperparameters used for the RL algorithms are not specified in the text.
> > The architectural details for any of the evaluations (unsupervised or RL) are also not described in the text.
>
> We thought it was best to commit these in the code; the code was given to the reviewers and the repo be open sourced after review.

---

> > ### Comment · Reviewer_gxYx · 2021-09-29
> > **The paper is much improved**
> >
> > I thank the authors for their clarifying comments and paper edits. The majority of my concerns have been addressed. Specifically, the authors have added an extensive hyperparameter search, created simulated robotics environments, added further explanation of how this benchmark differs from other deformable benchmarks (varied topologies), and demonstrated the complexity of this task set. I've updated my score accordingly.

---

> > ### Author Response · Authors · 2021-09-29
> > **Differences from existing task suites for deformables**
> >
> > > It seems that there are already existing RL benchmarks for deformable objects.
> > > it's not clear what interesting challenges these novel objects pose to RL algorithms that aren't already in existing deformable benchmarks such as SoftGym.
> >
> > The first key difference is that providing a fixed immutable set of benchmarks is not our main goal. After the discussions in the recent workshops on deformable objects manipulation and simulation (at IROS2019 [r11], ICRA2020 [r12], RSS2021 [r13]), it became clear that the community needs flexible suites of tasks (import custom objects, adjust simulation parameters and scenes easily). Researchers in computer vision and representation learning need visual variety (rather than physical flexibility). Our task suite provides these two key aspects. Table1 in Section2 outlines the current configurability/flexibility (or lack thereof) of the existing benchmarks/suites (last column). The next-to-last column notes the lack of object variability in existing suites (1-10 objects for deformables). In contrast, we provided 130+ built-in mesh variants, added support for procedural generation, and demonstrated loading more than 1000 additional meshes from external sources, with varying textures and material parameters. Such a simulation task suite for highly deformable objects has not been available before.
> >
> > The second key difference is that for this suite we focused on highly deformable objects with non-trivial topology and geometry. Our environments can be easily extended to include more of the thick and slightly deformables objects (similar to our hoop example). However, as we outlined in our response to reviewer3 (above) -- highly deformable objects present a unique set of challenges, both for state representation and for simulation. For researchers who prefer to consider simpler topologies and simpler geometric shapes and are seeking a benchmark with a fixed set of environments -- existing suites, such as SoftGym, provide a very good option. SoftGym focuses on ropes and rectangular cloth which are definitely important as primitive objects. However, a benchmark that analyzes the ability to capture the state of a rectangular cloth and manipulate it on a tabletop is not sufficient to cover manipulation with all kinds of highly flexible objects. For example, the success of manipulating garment-like objects with non-trivial topology often hinges on the ability to solve a much more difficult state representation problem, both for the deformable object itself and for its relation to the environment. Discerning the topological features of these objects, such as openings, holes and connections becomes key, but these features are completely missing from the basic linear and rectangular shapes.
> >
> > > Authors claim that their platform makes it easier to add custom objects (you have to recompile SoftGym), but this seems like a minor technical difference.
> >
> > Thanks for bringing further attention to this. Loading a custom mesh into SoftGym requires a non-trivial step of figuring out how to specify mesh vertices and which parts of the code to update. We are not aware of the ability to customize texture of deformables in SoftGym+Flex. With DEDO+PyBullet, users can simply provide a path to .obj, texture file and initial object pose via a flag. Custom re-compilation involves more setup overhead that a 'pip-install'. For labs focusing on research in deformable object manipulation some overhead can be acceptable. We would like to enable collaboration with a wide variety of researchers - e.g. in topology, geometry, computer vision, and for these the simulation setup overhead could be unacceptable. For the track on open-sourcing datasets/benchmarks/environments - we thought that noticeably improving ‘user experience’ was a non-negligible factor.
> >
> > It is also worth noting that our benchmark poses perception challenges at a different level. Given the task relevance, identifying and locating above topological features is important [r14, r15]. Task states of a rope or a rectangular cloth could be well encoded by a top-view image [r16, r17], which was shown to be quite effective in solving SoftGym tasks [r18]. In contrast, perceiving garment-like objects is still an open question [r19, r20] and learning a specific dressing skill requires a substantial reward engineering effort, even when the policy can access the state ground truth [r21]. To this end, our benchmark provides rich texture variations and customizability to facilitate visual learning, which is also something that is not present in SoftGym and other existing suites with highly deformable objects.
> >
> > REFERENCES (in the comment below)

---

> > > ### Author Response · Authors · 2021-09-29
> > > **REFERENCES for "Differences from existing task suites for deformables"**
> > >
> > > [r11] Panel discussion of the DO-Sim: Workshop on Deformable Object Simulation in Robotics at the Robotics: Science and Systems Conference (RSS) 2021. https://sites.google.com/nvidia.com/do-sim
> > >
> > > [r12] Workshop on Representing and Manipulating Deformable Objects. ICRA 2021. See open discussion round 2 at https://deformable-workshop.github.io/icra2021/#schedule
> > >
> > > [r13] Workshop on Robotic Manipulation of Deformable Objects. IROS 2020. See panel discussion at http://commandia.unizar.es/irosworkshop2020/
> > >
> > > [r14] Koganti et al, Bayesian Nonparametric Learning of Cloth Models for Real-Time State Estimation, IEEE Transactions on Robotics 2017
> > >
> > > [r15] Twardon and Ritter, Interaction skills for a coat-check robot: Identifying and handling the boundary components of clothes, ICRA 2015
> > >
> > > [r16] Sundaresan et al, Learning Rope Manipulation Policies Using Dense Object Descriptors Trained on Synthetic Depth Data, ICRA 2020
> > >
> > > [r17] Seita et al, Deep Transfer Learning of Pick Points on Fabric for Robot Bed-Making, ISRR 2019
> > >
> > > [r18] Ma et al, Learning Latent Graph Dynamics for Deformable Object Manipulation, arXiv:2104.12149, 2021
> > >
> > > [r19] Li et al, Recognition of deformable object category and pose, ICRA 2014
> > >
> > > [r20] Sun et al, Autonomous Clothes Manipulation Using a Hierarchical Vision Architecture, IEEE Access 2018
> > >
> > > [r21] Clegg et al, Learning to Dress: Synthesizing Human Dressing Motion via Deep Reinforcement Learning, ACM Transactions on Graphics 2018

---

### Official Review · Reviewer_1hWZ · 2021-09-19
**Review for Dynamic Environments with Deformable Objects**

**Rating:** 7
**Confidence:** 3
**Clarity:** The paper is clearly written.

**Strengths:**

The paper is very well written, each task is explained in detail. The code and documentation are also very comprehensive.
The environment presented has some strong points:
1. Variety of 3d models of soft bodies and textures
2. Ease of use
3. Authors show benchmark results on these tasks using widely used RL algorithms like PPO and SAC

**Weaknesses:**

I am a bit concerned with how similar all tasks are in terms of executing them. For example, hang garment task and buttoning cloth seems to require the same kind of skill. Similarly, hang a mask, hang a backpack, and put a dress garment seems to also require the same set of skills. Most of these tasks could be grouped under the same category with slight variation in them for generalization. I feel this is not a comprehensive environment for training on deformable objects in general but rather a training on a specific type of deformable object. I am interested to know if authors have thought about including few other scenarios that involved soft bodies in some other context.

**Additional Feedback:**

None

**Correctness:**

To the best of my knowledge, the implementation of the environment sounds correct although I am not an expert in 3d env development.

**Documentation:**

The paper is missing a link to a website that has documentation on environment specifics and the code

**Ethics:**

I agree with the authors in short term there are no ethical concerns and long-term effects are much broader and out of the scope of this work

**Relation To Prior Work:**

The paper cites a number of environments that support deformable objects.

**Summary And Contributions:**

The paper presents benchmark tasks that focus on interaction involving soft bodies or deformable-body. The paper presents a task divided into 4 types hanging garment by hook, put cloth holes through a button, throwing hoops and lasso around the pole, and putting the garment on a mannequin. The environment is based on Pybullet with is open source and supports customization and addition of external 3d models. The paper also presents benchmarking results on these tasks

---

> ### Author Response · Authors · 2021-09-29
> **A detailed response regarding task difficulty/similarity**
>
> Tasks with highly deformable objects typically have a much larger state space than in cases with rigid or slightly deformable objects. Tasks that might appear semantically similar, such as ‘dress with a garment’ and ‘put on a backpack’, involve very different skills due to the deformation properties (and momentum), elasticity and task requirements. For instance, humans could learn to exploit the momentum of a backpack to hang it or put it on, but this is rarely the case for masks, where the passive dynamics is not significant. Under the same reward design, the RL performance also generally varies across these semantically similar tasks, implying different perceptual and dynamical challenges for policies to bring the objects towards the goal.
> In robotics, generalizing category-level skills is quite a recent development even for rigid objects [1]. In computer vision, the state of the art methods struggle to generalize across different objects and tasks [8]. Research on deformable manipulation is often limited to one or a few instances of a specific object category. Typical examples include (in separate publications): manipulating an elastic loop [2]; putting on a hat, shirt [4], gown [5], sleeveless jacket [6]; hanging cloth [7]. Our benchmark covers these representative skills that pertain to existing challenges (interests from robotics, graphics, computer vision), while providing extensive variations and customizability to facilitate further research.
> To illustrate the fact that the same RL algorithm can have a vastly different performance on these tasks, we included new analysis in the new Section 6 (page 10) in the main paper, and provided new additional plots in the supplementary. It would be easy to import a wide variety of deformable objects into our suite, since PyBullet can import .obj files with the mesh data. However, one could argue that tasks with objects that are only slightly deformable can already benefit from existing research for rigid objects. For example, the recently popular keypoint extraction methods can generalize to the case of objects that are somewhat deformable but still mostly maintain their shape (e.g. plush toys [9], flexible shoes [10]). However, most of these algorithms would not be applicable to the case of highly deformable fabrics, ropes and cables, where the object does not ever return to a canonical shape during manipulation. Such cases could benefit from new representation learning techniques and from new structured representation insights drawn from various research communities. Hence, while such tasks might involve ‘just’ thin-shell objects made from fabrics and flexible materials, learning to represent and manipulate such objects reliably (and without known models or hard-coded assumptions) is beyond the current state of the art. Hence, we focus on this area.
>
> REFERENCES:
>
> [1] Gao and Tedrake, kPAM 2.0: Feedback Control for Category-Level Robotic Manipulation, IEEE RA-L 2021
>
> [2] Yoshida et al, Simulation-based optimal motion planning for deformable object, IEEE ARSO 2015
>
> [3] Klee et al, Personalized assistance for dressing users, International Conference on Social Robotics, 2015
>
> [4] Clegg et al, Learning to Dress: Synthesizing Human Dressing Motion via Deep Reinforcement Learning, ACM Transactions on Graphics 2018
>
> [5] Kapusta et al, Personalized collaborative plans for robot-assisted dressing via optimization and simulation, Autonomous Robots 2019
>
> [6] Li et al, Provably Safe and Efficient Motion Planning with Uncertain Human Dynamics, RSS 2021
>
> [7] Matas et al, Sim-to-Real Reinforcement Learning for Deformable Object Manipulation, CoRL 2018
>
> [8] Li, Yunzhu, et al. "Causal Discovery in Physical Systems from Videos." Advances in Neural Information Processing Systems 33 (2020)
>
> [9] Florence et al. “Dense Object Nets: Learning Dense Visual Object Descriptors By and For Robotic Manipulation”, CoRL 2018
>
> [10] Manuelli et al. “kPAM: KeyPoint Affordances for Category-Level Robotic Manipulation”, ISRR 2019

---

> > ### Author Response · Authors · 2021-09-29
> > **A connection with other responses regarding DEDO coverage/versatility**
> >
> > Reviewer4 (the last response) also initially expressed concerns that tasks might be not sufficiently varied and perhaps too easy. This motivated us to show that the tasks span a wide range of levels of difficulty. To avoid copy/pasting, we would appreciate if other reviewers with such concerns could take a look at this thread that provides our responses: https://openreview.net/forum?id=WcY35wjmCBA&noteId=nedljHWiwkD
> >
> > In the updated supplementary pdf we show that PPO solves Button-v1 task. This is likely because the trajectory for this task requires a direct motion towards the buttons, while cloth being attached to a support structure prevents the policy from overshooting too far. In contrast, the seemingly ‘easy’ HangGarment-v1 task proves challenging (as we show on the new page 10). This could be because the motion is less restricted and small changes in the motion can result in missing the hook entirely. DressGarment-v1 task proves intractable to the leading RL algorithms even after an extensive hyperparameter search. It may seem that these tasks have similar aspects, such as loops/holes that need to be put on hooks/arms. However, the differences in the task setup give rise to large variations in complexity. There is room for setting up the tasks where basic perception is enough, versus tasks where there is a crucial need to track small+thin parts of objects to succeed, versus tasks that are challenging because there are only a few ways to fit the deformable object onto a complex rigid object. With DEDO we aim for a flexible set of environments with dynamic tasks and highly deformable objects, because these aspects alone can provide a rich variability in task complexity. Moreover, many of these tasks can be intractable to the existing computer vision and control approaches: vision is challenging, because object shape changes rapidly; control is challenging, because there is usually little force feedback and no low-dimensional canonical representation for the state. Hence, solving DEDO tasks would require a significant step forward in both vision/perception and control/policy learning.
> >
> > Though if the reviewers feel that we can add a particular type of task that is missing - we can surely work on such an extension. Our suite was specifically designed to allow easy new task construction.

---

> > > ### Comment · Reviewer_1hWZ · 2021-09-30
> > > **Response to rebuttal**
> > >
> > > My concerns have been fully answered. I am increasing the paper rating accordingly.

---

> > > > ### Author Response · Authors · 2021-09-30
> > > > **Added an additional customization example in the supplementary**
> > > >
> > > > Thank you for taking the time to read our responses. We are glad that we were able to clarify the primary goals for the DEDO suite and address the points regarding the task difficulty and diversity.
> > > >
> > > > > I am interested to know if authors have thought about including few other scenarios that involved soft bodies in some other context.
> > > >
> > > > While with DEDO we wanted to fully focus on tasks with highly deformable objects that have interesting  topologies, the review comments helped us see that there could be other directions where DEDO could be beneficial. Hence, we included a short example in the supplementary of how to construct a FoodPacking task, with food objects from the YCB dataset : cans and food packages loaded as rigid objects, fruit objects as deformables.

---

> ### Author Response · Authors · 2021-09-29
> **Missing a link to a website clarification**
>
> The organizers instructed us that during the review process we can provide a private link that is visible to the reviewers. We provided this link as part of the original submission. It includes our code repository, a detailed README, and documentation wiki. We hope that the organizers can clarify to the reviewers where this link can be found in the reviewer’s private view. We can see this link on our side under the “Private Dataset URL”, but we do not know why some of the reviewers were not able to see it on their side. Other reviewers did see this link and made comments regarding our code repository, indicating that all the data was accessible to them. Please let us know if you cannot see this information and we will contact the organizers to get their help with this matter.

---

### Official Review · Reviewer_a4KA · 2021-09-19
**A suite of benchmarks for the 3D manipulation and control of deformable objects**

**Rating:** 7
**Confidence:** 3
**Correctness:** The claims appear to be correct.
**Clarity:** The paper is reasonably well written …

**Strengths:**

--The authors provide a suite of 3D manipulation benchmarks for tasks with deformable objects. This type of benchmark is quite hard to make since it requires an understanding of graphics and physics engines. Therefore, this will allow researchers to focus on generating control policies.
--The code is accessible and is reasonably extensible, with particular strength in bringing in custom assets.

**Weaknesses:**

--It is unclear if these simulators are sufficient for sim2real transfer of policies or unsupervised learning approaches. Therefore, the usefulness of the simulators is limited to understanding how to build policies for complex control problems and the link between control policies and observation representations.
--The paper would benefit from a clear statement, via a table, of the runtime speeds of the suite of simulators.
--The lack of a code testing suite will make maintaining the benchmarks challenging and limit the extensibility of the benchmarks.

**Additional Feedback:**

The paper lacks a titled section that discusses the limitations and scope of the benchmarks.

**Documentation:**

The authors have provided well-documented code with examples and reference implementations.

**Relation To Prior Work:**

The references to prior work appear adequate.

**Summary And Contributions:**

In this paper, the authors present a set of environments for reinforcement learning for control theory. These environments focus on the manipulation and movement of deformable objects in 3D space. The environments are simulated with PyBullet and equipped with an OpenAI gym interface. The authors also provide some reference implementations of unsupervised representation learning methods and trained several RL algorithms using well-known open-source packages.

The paper's main contribution is a set of new environments for 3D manipulation and control of deformable objects. The authors provide a wide range of object topologies and procedural materials with a suite of tasks like hanging bags and buttoning cloth. Additionally, the environments come with customizability and extension to adjust to a user's given goals.

---

> ### Author Response · Authors · 2021-09-29
> **sim2real considerations**
>
> > It is unclear if these simulators are sufficient for sim2real transfer of policies or unsupervised learning approaches
>
> In CoRL (a leading venue devoted to learning-for-robotics) a paper titled “Sim-to-Real Reinforcement Learning for Deformable Object Manipulation” showed transfer with a real robot manipulating a rectangular piece of red cloth, using PyBullet simulator engine. So there is definitely hope, but as a research community we need to be carefully optimistic about this. Certain aspects could facilitate transfer - e.g. ability to texturize the simulated environment to very closely resemble the real one could go a long way to make vision-based parts transfer well. However, various lightning and camera noise/distortions could still leave a visual sim-to-real gap. Some aspects of physics modeling are also beyond the state-of-the-art: e.g. fine interactions between a strap of cloth and a rigid gripper is not something that can be fast, physically plausible and visually exact all at the same time now.
>
> Hence, with DEDO our take on furthering the sim-to-real efforts is similar to that shown to be effective with the AI Habitat. AI Habitat is a large-scale simulation suite with a fast renderer. Despite favoring speed over fidelity (e.g using a low simulation frequency), “Sim2Real predictivity” work by Kadian et al. showed that evaluating methods trained in AI Habitat correlated well with how well these methods performed when learning from real data. Hence, for a simulation suite to be useful for real-world efforts, it is not always strictly necessary that the environments themselves can be exactly aligned to a particular real-world situation. Nonetheless, it is crucial that environments provide similar kinds of challenges and complexity, i.e. it is important that they do not oversimplify the problem. With that, we can use DEDO suite to first judge the potential of the learning methods on simulated data. Then, the promising data efficient methods can be tested with learning from real data. This can be done without mandating a direct sim-to-real setup, and instead  using simulation to filter out methods that are not likely to succeed on real data.
>
> With DEDO we took care to ensure that some of our tasks are tractable, while others are very challenging. The simpler tasks could be solved with existing methods without reward engineering or domain knowledge. These tasks could be set up for sim-to-real, if the community decides that this is an interesting direction. The majority of DEDO tasks, however, are beyond the state of the art both for vision and for control (for methods that do not make use of particular task-specific domain knowledge). For such challenging tasks even finding a method that performs well in simulation would be non-trivial. So DEDO can facilitate this aspect of using simulation to gauge the potential of the novel approaches.
>
> REFERENCES:
>
> J Matas et al. “Sim-to-Real Reinforcement Learning for Deformable Object Manipulation”, CoRL 2018
>
> Kadian et al. “Sim2Real predictivity: Does evaluation in simulation predict real-world performance?” IEEE Robotics and Automation Letters. 2020.

---

### Official Review · Reviewer_tywg · 2021-09-21
**A solid contribution to promote research on deformable objects**

**Rating:** 7
**Confidence:** 3

**Strengths:**

+ Addresses a need articulated by the robotics community (customizable environments for studying deformable objects)
+ More diversity and customizability than prior such environments
+ Clear presentation and extensive documentation in the code repo

**Weaknesses:**

- The pipeline for customizing action and observation spaces, and reward functions, is not described in the main paper
- The VAE and RL experiments are not especially illuminating

-- Customizability --

I think the focus on customizability is great, and deserves a bit more detail. For example: does the platform support observation spaces that are not pixels? How would a user go about specifying a new action space or reward function? Can the fidelity of the simulation be changed based on user demands?

-- VAE and RL experiments --

The VAE experiments don’t fully describe the evaluation metric. From the figure I can infer that it is reconstruction error. Yet the text alludes to the fact that VAEs might learn a structured latent space and the latent representation could be useful for tracking holes, etc. I think it would be more interesting to investigate the VAE variants with respect to the latent structure they learn, rather than with respect to their ability to reconstruct test images. RL on top of VAE embeddings could also be interesting.

In general, both the VAE and RL experiments serve as a good demonstration of how the platform might be used, but do not yield many new insights into unsupervised learning or RL with deformable objects. I think that's okay, since that's not the focus of the paper, but perhaps could be improved.



**Additional Feedback:**

None

**Clarity:**

The writing is quite clear overall.

The figures are also helpful, but could be improved. Many of the figures are raw screenshots without annotation. I think it could help to add more annotation, labeling different parts of the visualizations and pointing out qualitative things to look at. Figure 2 is a good example of what I'm thinking of; that kind of markup could be added to other figures as well.

Figures 11 and 12 show raw TensorBoard screenshots, which I don't think are very helpful. I take it that the intent is to show how the system is integrated with TensorBoard, but this is so standard that I think just stating it in text is enough. To me it would be better to use the space to show interesting results of the experiments, without the distraction of TensorBoard details.

Details on some of the tasks are missing. For example, the mathematical form of the actions space and reward function is only given for some tasks. I also could not easily find these details in the wiki. This information should be added somewhere in text, so that users don't have to hunt through the code.

Minor comments:
- “DEDO” is referred to, in Table 1, before being defined, in Section 3. Also, Table 1 is not referred to in the main text.
- Sewing-v0: why not select from all 22000 meshes? Why only 1200?
- Line 273: “researches” —> “researchers”

**Correctness:**

All claims appear correct to me: as far as I can tell, the dataset is constructed in a sound way.

**Documentation:**

The dataset is mostly well documented in the paper and code repo. In the "Clarity" section above, I commented on some missing details that should be added.

How the meshes, textures, and other assets were collected is not fully described. I see that some are procedurally generated and others can be sampled from existing databases. However it seems that some assets are human-designed and new to this paper. The process for creating those should be described.

**Ethics:**

I do not have concerns.

**Relation To Prior Work:**

The relationship to prior work is well discussed. Table 1 is very helpful.

**Summary And Contributions:**

This paper describes a new simulation platform for studying dynamic manipulation of deformable objects. The platform is built on top of PyBullet, and consists of a suite of tasks involving different object categories and manipulation goals. The suite contains more task and object types than prior environments for deformable objects, and is built to be customizable. Initial experiments demonstrate how the platform can be used for representation learning and reinforcement learning research.

---

> ### Author Response · Authors · 2021-09-29
> **Customizability details**
>
> > The pipeline for customizing action and observation spaces, and reward functions, is not described in the main paper
> > does the platform support observation spaces that are not pixels?
>
> Judging from the current excitement about learning from pixels we thought that the community would be primarily interested in that representation, hence for this submission with focused on RGB images.
> In addition, we do provide a low-dimensional observation space that reports the position/velocity of the grippers/anchors, and in the newly added version for dual arm manipulation we report pose/velocity of the end-effectors.
> Switching to this low-dimensional representation/observation space can be done by setting --cam_resolution flag to 0.
> Mesh vertex position can be also included as observations - such representation would include information about the state of the deformables, while abstracting away other complexities, such as appearance and occlusions.
>
> > How would a user go about specifying a new action space or reward function?
>
> Specifying a new reward can be done by overriding get_reward() function in DeformEnv class.
> For action spaces: DEDO uses velocity control for gripper anchors and Cartesian (end-effector pose) control for the robots. Our newly added BulletManipulator class also provides joint position, velocity and torque control interfaces. We aimed to make DEDO simple to use for non-robotics users, so we implemented a wrapper function that uses inverse kinematics to solve for the joint commands given desired 3D/Cartesian gripper poses. But if there is interest, we can easily expose all the joint-level position/velocity/torque control options via a flag.
>
> > Can the fidelity of the simulation be changed based on user demands?
>
> The value for --sim_freq flag can be increased/decreased to provide higher/lower fidelity and hence the fidelity-vs-speed tradeoff.
>
> Simulation fidelity-vs-speed will also depend on mesh density, and one easy way to expose this would be to offer --mesh_subsampling_level flag. We did not think of this initially, but now this suggests to us a very interesting direction. The mesh can be adaptively re-loaded after every episode reset. Using such flag, users could experiment with advanced learning approaches that vary the mesh fidelity adaptively during training. We can add this functionality right away for procedurally generated meshes; we also can look into enabling this for existing meshes provided via .obj files.

---

> > ### Comment · Reviewer_tywg · 2021-09-30
> > **Review update**
> >
> > This all sounds good as well -- I think adding more of this discussion to the main paper will help the final version.
> >
> > I think adding the --mesh_subsampling_level flag could be really useful! Perhaps low fildelity simulation is sufficient for training certain policies, or, as the authors mention, the fidelity could be annealed over training iterations or dynamically, which would be really interesting. This would open up a lot of fun possibilities and might make the platform easier to use as well.
> >
> > Based on the author responses and other reviews, I maintain my opinion that this is strong work and well above the bar for acceptance.

---

> ### Author Response · Authors · 2021-09-29
> **VAE/RL and new directions**
>
> > In general, both the VAE and RL experiments serve as a good demonstration of how the platform might be used, but do not yield many new insights into unsupervised learning or RL with deformable objects. I think that's okay, since that's not the focus of the paper, but perhaps could be improved.
>
> Indeed, we aimed to stay away from suggesting concrete novel research directions in this submission to avoid turning it into a “new method” paper. Hence the focus for VAE/RL section was to show examples of what the existing unsupervised/RL can learn from the DEDO environments. The main point of the RL section was to show interoperability with two leading RL frameworks (one user-friendly - StableBaselines3, one for large-scale RL - RLlib).
>
> > The VAE experiments don’t fully describe the evaluation metric. From the figure I can infer that it is reconstruction error. Yet the text alludes to the fact that VAEs might learn a structured latent space and the latent representation could be useful for tracking holes, etc.
>
> VAE approaches we showed use variations of ELBO losses during training, which are not directly comparable to each other. Hence, the reconstruction part of the loss is the only default common/comparable quantity.
> While reconstruction quality as an evaluation metric is ubiquitous in ML literature, we do agree that this is not an ideal way to judge whether a method has learned a useful latent state. To elaborate: good reconstructions are needed for the gradients to flow well through the decoder back to the latents and the encoder; however, reconstruction loss may not be the final arbiter for the quality of the latent state.
>
> > I think it would be more interesting to investigate the VAE variants with respect to the latent structure they learn, rather than with respect to their ability to reconstruct test images
>
> That is an interesting point that should be easy to incorporate into DEDO. Since we have ground truth for the holes/openings in the deformables -- we can easily evaluate unsupervised approaches based on whether their latent state is enough to reconstruct the hole locations. Our guess is that many existing unsupervised methods will struggle with this, despite being successful in other ways on other datasets frequently used in the ML/vision literature. In one of our previous projects we developed a set of evaluation criteria to uncover whether unsupervised learning approaches capture the state of rigid objects appropriately. We could port this code to DEDO and evaluate whether the key topological or geometric aspects of the state are present in the learned latent representations. We thank the reviewer for guiding us toward this realization, and will work on adding this evaluation metric to DEDO for approaches that learn latent representations.

---

> > ### Comment · Reviewer_tywg · 2021-09-30
> > **Review update**
> >
> > Thanks for addressing these concerns. I agree that it's fine to leave deeper investigations of unsupervised learning and RL to other papers (which can use this dataset!).
> >
> > But I'm also happy to hear the interest in porting latent topological/geometric evaluation into DEDO. I think this could be very interesting and look forward to seeing how it pans out!

---

### Decision · Program_Chairs · 2021-10-09

**Decision:**

Accept

**Comment:**

All reviewers agree on acceptance. I recommend the authors to take into account the reviewers' comments to improve the paper for the camera-ready version.